# Statistical Theory of Differentially Private Marginal-based Data Synthesis Algorithms

**Ximing Li**
Tsinghua University
Beijing, 100084, P. R. China
`{li-xm19}@mails.tsinghua.edu.cn`

**Chendi Wang**
Shenzhen Research Institute of Big Data &
University of Pennsylvania
Philadelphia, PA 19104, USA
`{chendi}@wharton.upenn.edu`

**Guang Cheng**
Department of Statistics
University of California, Los Angeles
Los Angeles, CA 90095, USA
`{guangcheng}@ucla.edu`

## Abstract

Marginal-based methods achieve promising performance in the synthetic data competition hosted by the National Institute of Standards and Technology (NIST). To deal with high-dimensional data, the distribution of synthetic data is represented by a probabilistic graphical model (e.g., a Bayesian network), while the raw data distribution is approximated by a collection of low-dimensional marginals. Differential privacy (DP) is guaranteed by introducing random noise to each low-dimensional marginal distribution. Despite its promising performance in practice, the statistical properties of marginal-based methods are rarely studied in the literature. In this paper, we study DP data synthesis algorithms based on Bayesian networks (BN) from a statistical perspective. We establish a rigorous accuracy guarantee for BN-based algorithms, where the errors are measured by the total variation (TV) distance or the $L^2$ distance. Related to downstream machine learning tasks, an upper bound for the utility error of the DP synthetic data is also derived. To complete the picture, we establish a lower bound for TV accuracy that holds for every $\epsilon$-DP synthetic data generator.

## 1 Introduction

In recent years, the problem of privacy-preserving data analysis has become increasingly important and *differential privacy* (Dwork et al., 2006) appears as the foundation of data privacy. Differential privacy (DP) techniques are widely adopted by industrial companies and the U.S. Census Bureau (Johnson et al., 2017; Erlingsson et al., 2014; Nguyên et al., 2016; The U.S. Census Bureau, 2020; Abowd, 2018).

One important method to protect data privacy is differentially private data synthesis (DPDS). In the setting of DPDS, a synthetic dataset is generated by some DP data synthesis algorithms from a real dataset. Then, one can release the synthetic dataset and the real dataset will be protected. Recently, National Institutes of Standards and Technology (NIST) organized the differential privacy synthetic data competition (NIST, 2018; 2019; 2020-2021). In the NIST competition, the state-of-the-art algorithms are marginal-based (McKenna et al., 2021), where the synthetic dataset is drawn from a noisy marginal distribution estimated by the real dataset. To deal with high-dimensional data, the distribution is usually modeled by the probabilistic graphical model (PGM) such as the Bayesian networks or Markov random fields (Jordan, 1999; Wainwright et al., 2008; Zhang et al., 2017; Mckenna et al., 2019; Cai et al., 2021).

Despite its empirical success in releasing high-dimensional data, as far as we know, the theoretical guarantee of marginal-based DPDS approaches is rarely studied in literature. In this paper, we focus on a DPDS algorithm based on the Bayesian networks (BN) known as the PrivBayes (Zhang et al.,

2017) that is widely used in synthesizing sparse data (sparsity measured by the degree of a BN that will be defined later). A BN is a directed acyclic graph where each vertex is a low-dimensional marginal distribution and each edge is the conditional distribution between two vertices. It approximates the high-dimensional distribution of the raw data with a set of well-chosen low-dimensional distributions. Random noise is added to each low-dimensional marginal to achieve differential privacy. We aim to analyze the marginal-based approach from a statistical perspective and measure the accuracy of PrivBayes under different statistical distances including the total variation distance or the $L^2$ distance.

Another metric of synthetic data we are interested in is the utility metric related to downstream machine learning tasks. Empirical evaluation of synthetic data in downstream machine learning tasks is widely studied in literature. Existing utility metrics include Train on Synthetic data and Test on Real data (TSTR, (Esteban et al., 2017)) and Synthetic Ranking Agreement (SRA, (Jordon et al., 2018)). To our best knowledge, most of these utility evaluation methods are empirical without a theoretical guarantee. Establishing the statistical learning theory of synthetic data is another concern of this paper. Precisely, we focus on the statistical theory of PrivBayes based on the TSTR error.

**Our contributions.** Our contributions are three-fold. First, we theoretically analyze the marginal-based synthetic data generation and derive an upper bound on the TV distance and $L^2$ distance between real data and synthetic data. The upper bounds show that the Bayesian network structure mitigates the "curse of dimensionality". An upper bound for the sparsity of real data is also derived from the accuracy bounds. Second, we evaluate the utility of the synthetic data from downstream supervised learning tasks theoretically. Precisely, we bound the TSTR error between the predictors trained on real data and synthetic data. Third, we establish a lower bound for the TV distance between the synthetic data distribution and the real data distribution.

## 1.1 RELATED WORKS AND COMPARISONS

Broadly speaking, our work is related to a vast body of work in differential privacy (Dinur & Nissim, 2003; Dwork & Nissim, 2004; Blum et al., 2005; Dwork et al., 2007; Nissim et al., 2007; Barak et al., 2007; McSherry & Talwar, 2007; Machanavajjhala et al., 2008; Dwork et al., 2015). For example, McSherry & Talwar (2007) proposed the exponential mechanism that is widely used in practice. Machanavajjhala et al. (2008) discussed privacy for histogram data by sampling from the perturbed cell probabilities. However, these methods are not efficient for releasing high-dimensional tabular data, since the domain size grows exponentially in the dimension (which is known as "the curse of dimensionality"). The state-of-art method for this problem is the marginal-based approach (Zhang et al., 2017; Qardaji et al., 2014; Zhang et al., 2021). Zhang et al. (2017) approximated the raw dataset by a sparse Bayesian network and then added noise to each vertex in the graph. Zhang et al. (2021) selected a collection of 2-way marginals and a gradually updating method was applied to release synthetic data. Although most of them provide rigorous privacy guarantees, theoretical analysis on accuracy is rare. Wasserman & Zhou (2010) established a statistical framework of DP and derived the accuracy of distribution estimated by noisy histograms. Our setting is different from theirs. Precisely, we analyze how noise addition and post-processing affect the conditional distribution (Lemma 6.2). Moreover, our proof handles the non-trivial interaction between the Bayesian network and noise addition.

Our lower bound (Theorem 5.1) is related to existing results of the worst case lower bounds under the DP constraint in literature (Hardt & Talwar, 2010; Ullman, 2013; Bassily et al., 2014; Steinke & Ullman, 2017). Hardt & Talwar (2010) established lower bounds for the accuracy of answering linear queries with privacy budget $\epsilon$. Ullman (2013) derived the worst-case result that in general, it is NP-hard to release private synthetic data which accurately preserves all two-dimensional marginals. Bassily et al. (2014) built on their result and further developed lower bounds for the excess risk for every $(\epsilon, \delta)$-DP algorithm. Our result is novel since we consider private synthetic data and the corresponding TV accuracy. Existing results for linear quires are not directly applicable to TV accuracy since they heavily rely on the linear structure.

## 2 DIFFERENTIAL PRIVACY

Differential privacy requires that any particular element in the raw dataset has a limited influence on the output (Dwork et al., 2006). The definition is formalized as follows. Here the data domain is denoted as $\Omega$.

**Definition 2.1** (($\epsilon, \delta$)-differential privacy). *Let $\mathcal{A} : \Omega^n \to \mathcal{R}$ be a randomized algorithm that takes a dataset of size $n$ as input, where the output space $\mathcal{R}$ is a probability space. For every $\epsilon, \delta \geq 0$, $\mathcal{A}$ satisfies ($\epsilon, \delta$)-differential privacy if for every two adjacent datasets $D_1$ and $D_2$, we have*

$$\mathbb{P}[\mathcal{A}(D_1) \in S] \leq \exp(\epsilon)\mathbb{P}[\mathcal{A}(D_2) \in S] + \delta, \qquad \text{for all measurable } S \subseteq \mathcal{R}.$$

*Here $D_1$ and $D_2$ are datasets of size $n$. We say that they are adjacent if they differ only on a single element, denoted as $D_1 \simeq D_2$.*

For $\delta = 0$, we abbreviate the definition as $\epsilon$-differential privacy ($\epsilon$-DP). A widely used meta-mechanism to ensure $\epsilon$-DP is the Laplace mechanism. The Laplace mechanism privatizes a function $f$ on the dataset $D$ by adding i.i.d. Laplace noises (denoted as $\eta \sim \text{Lap}(\lambda)$ ) to each output value of $f(D)$. Here the probability density function of $\eta$ is given by $\mathbb{P}[\eta = x] = \frac{1}{2\lambda} \exp(\frac{-|x|}{\lambda})$. Dwork & Nissim (2004) show that it ensures $\epsilon$-DP when $\lambda \geq \Delta_f/\epsilon$, where $\Delta_f$ is the $L^1$ sensitivity of $f$:

$$\Delta_f = \max_{(D_1, D_2): D_1 \simeq D_2} \|f(D_1) - f(D_2)\|_1.$$

## 3 MARGINAL-BASED DATA SYNTHESIS ALGORITHMS

In this section, we introduce DP marginal-based methods. For simplicity, we consider Boolean data where $\Omega = \{0, 1\}^d$ and $|\Omega| = 2^d$. It's obvious that our theory can be generalized to any categorical dataset with a finite domain size.

### 3.1 DIFFERENTIALLY PRIVATE ESTIMATE OF LOW-DIMENSIONAL MARGINAL DISTRIBUTIONS

Given a dataset $D = \{x^{(i)}\}_{i=1}^n \subset \Omega^n$ drawn independently from a distribution, the probability mass function is estimated by

$$p_D(x) = \frac{1}{n} \sum_{i=1}^n \mathbb{1}[x^{(i)} = x], \qquad \text{for all } x \in \Omega. \tag{1}$$

**Noise addition and post processing.** We then sanitize $p_D(x)$ by the Laplace mechanism. Note that the sensitivity of $p_D(x)$ is $1/n$. Then, we define $\widetilde{p}_D = p_D + \text{Lap}(1/(n\epsilon))$ and $\widetilde{p}_D$ is $\epsilon$-DP. Adding noise leads to inconsistency. To be specific, some estimated probabilities may be negative and the overall summation may not be 1. The following two kinds of post processing methods to address the inconsistency are widely adopted in marginal-based methods (cf., (Mckenna et al., 2019; Zhang et al., 2017)).

*Normalization.* We convert all the negative probabilities to zeros, and then normalize all the probabilities by a scalar such that their summation is 1.

$L^2$-*projection.* We project the inconsistent distribution onto the probability simplex using the $L^2$ metric. Specifically, for an inconsistent distribution $(a_1, \cdots, a_m)$, the output is

$$(b_1, \cdots, b_m) := \underset{\widetilde{b}_i \geq 0, \sum \widetilde{b}_i = 1}{\arg \min} \sum_{i=1}^m (a_i - \widetilde{b}_i)^2.$$

### 3.2 MARGINAL SELECTION AND BAYESIAN NETWORKS

It is well-known that marginal-based methods have the curse of dimensionality. One way to mitigate the curse of dimensionality is adopting Bayesian networks (Zhang et al., 2017).

**Marginal selection.** We first disassemble the raw dataset into a group of lower dimensional marginal datasets. Precisely, PrivBayes (Zhang et al., 2017) uses a sparse Bayesian network $\{x_1, \cdots, x_d\}$ to approximate the raw data. Each node $x_i$ corresponds to an attribute, and each edge from $x_j$ to $x_i$ represents $\mathbb{P}[x_i \mid x_j]$, which is the probability of $x_j$ causing $x_i$. We denote $\Pi_i := \{j \mid x_j \to x_i\}$, which is the collection of all the attributes that affect $x_i$. Zhang et al. (2017) also make the following assumptions on the network structure. Here $k$ is a pre-fixed parameter that is much smaller than $d$.

**Assumption 3.1** (Sparsity). *The degree of the Bayesian network is no more than $k$. Precisely, for any $i$, the size of $\Pi_i$ is no more than $k$.*

The second assumption ensures that the graph cannot contain loops, which aids sampling from the graph.

**Assumption 3.2.** *For any $i$, we have $\Pi_i \subset \{x_1, \cdots, x_{i-1}\}$.*

For example, the Bayesian network in Figure 1 satisfies Assumption 3.1 for $k = 2$ and Assumption 3.2. The joint distribution is $\mathbb{P}(5 \mid 4, 3)\mathbb{P}(4 \mid 3, 2)\mathbb{P}(3 \mid 2, 1)\mathbb{P}(2 \mid 1)$.

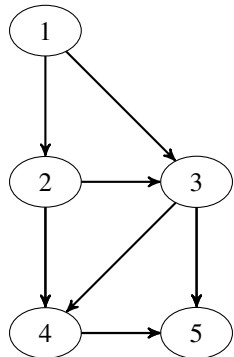

Figure 1: A Bayesian network over 5 attributes of degree 2

**DP Bayesian networks.** In a Bayesian network, each low-dimensional marginal distribution $\mathbb{P}(x_i, \Pi_i)$ is estimated by the marginal function defined by (1). For privacy consideration, we add Laplace noise $\mathrm{Lap}(d/n\epsilon)$ to the marginal $\mathbb{P}(x_i, \Pi_i)$ and obtain the DP distribution $\widehat{\mathbb{P}}[x_i, \Pi_i]$ by using post processing to the noisy marginal. Then the overall privacy budget can be calculated by the composition property of DP (Dwork, 2008; Zhang et al., 2017) and is $\epsilon$.

**Generating synthetic data.** The loop-free Bayesian network provides an efficient sampling approach. Precisely, we draw $x_i$ from $\widehat{\mathbb{P}}[x_i \mid \Pi_i]$ in an increasing order of $i$. Recall that Assumption 3.2 ensures that $x_j \notin \Pi_i$ for any $j > i$. Therefore, by the time $x_i$ is to be sampled, all nodes in $\Pi_i$ must have been sampled. This verifies that the sampling approach is practical. Moreover, sampling $x_i$ only needs the marginal $\widehat{\mathbb{P}}[x_i, \Pi_i]$, instead of the full distribution. By Assumption 3.1, it is a marginal with size less than $k + 1$. This leads to a small computational load since $k$ is small. With this sampling method, one (Zhang et al., 2017) can show that the synthetic private distribution is $\widehat{\mathbb{P}}[x_1, \cdots, x_d] = \prod_{i=1}^d \widehat{\mathbb{P}}[x_i \mid \Pi_i]$.

## 4 ACCURACY OF PRIVBAYES

In this section, we develop some theoretical results of the accuracy of PrivBayes. We discuss the proof of these results briefly in Section 6.

### 4.1 STATISTICAL DISTANCES

The goal of this subsection is to establish the accuracy guarantee for PrivBayes with different post-processing methods: normalization and $L^2$-projection. By the term "accuracy", we mean the TV or the $L^2$ distance between the synthetic distribution and the raw data distribution, respectively. Note that the error comes from two sources: 1) approximating the raw data by a Bayesian network that

satisfies Assumption 3.1 and Assumption 3.2 , 2) adding noise and post processing. The first error, however, only relies on the sparsity of the raw data. Since we aim to establish our result for general raw data, we only focus on the second one. Therefore, it is natural for us to make the following assumption.

**Assumption 4.1.** *We assume the raw data distribution can be represented by a Bayesian network with $d$ vertices that satisfies Assumption 3.1 and Assumption 3.2.*

With this assumption, PrivBayes (normalization) has the following accuracy guarantee.

**Theorem 4.1.** *Assuming that the raw dataset $\mathbb{D}$ is Boolean and satisfies Assumption 4.1, then we have*

$$\left\|\widehat{\mathbb{P}} - \mathbb{P}\right\|_{\mathrm{TV}} \leq \frac{12d^2 2^{2k}(k+1)}{n\epsilon} \log \frac{2d}{\delta},$$

*with probability at least $1 - \delta$ (with respect to the randomness of the Laplace mechanism and the same below). Here $\mathbb{P}$ is the empirical distribution of $\mathbb{D}$ and $\widehat{\mathbb{P}}$ is the output of PrivBayes (normalization) with privacy budget $\epsilon$.*

*Proof.* See Section 6 for a proof sketch. □

The other post-processing method ($L^2$-projection) we studied is also efficient. The following result verifies that PrivBayes ($L^2$-projection) enjoys the similar accuracy guarantee in terms of the $L^2$ distance. Here, with a little bit abuse of notations, we denote the $L^2$ distance between two distributions as the $L^2$ distance between their density functions.

**Theorem 4.2.** *Assuming the raw dataset $\mathbb{D}$ is Boolean and Assumption 4.1 is satisfied, then we have*

$$\left\|\widehat{\mathbb{P}} - \mathbb{P}\right\|_{L^2} \leq \frac{12d^2 2^k(k+1)}{n\epsilon} \log \frac{2d}{\delta},$$

*with probability at least $1 - \delta$. Here $\mathbb{P}$ is the empirical distribution of $\mathbb{D}$ and $\widehat{\mathbb{P}}$ is the output of PrivBayes ($L^2$-projection) with privacy budget $\epsilon$.*

*Proof.* The proof is similar to Theorem 4.1. However it is still non-trivial and we discuss it in detail in Appendix B. □

**Discussion.** Theorem 4.1 and Theorem 4.2 achieves a bound that is consistent (tends to 0 as $n$ tends to infinity) if $k \ll \log_2(n\epsilon/d^2)$. Moreover, a smaller $k$ leads to smaller upper bounds. Since we assume that the real data and synthetic data share the same $k$, we conclude that PrivBayes achieve better performance on sparser real datasets. The size of $k$ is often rather small in real application. For example, Zhang et al. (2017) chose $k \leq 4$ in the simulation. Theorem 4.1 also characterizes the reliance on the privacy budget $\epsilon$. Precisely, tighter privacy budget means better privacy guarantee, but leads to worse performance. Moreover, our rate is polynomial in the dimension $d$. Comparing with directly applying the Laplace mechanism to the whole domain (see Theorem 4.3), our result shows that by heavily deploying the network structure, the Bayesian network exponentially refines the rate.

**Theorem 4.3.** *Assuming that the raw dataset $\mathbb{D}$ is Boolean, then we have*

$$\left\|\widetilde{\mathbb{P}} - \mathbb{P}\right\|_{\mathrm{TV}} \leq \frac{d 2^{2d}}{n\epsilon} \log \frac{2}{\delta},$$

*with probability at least $1 - \delta$. Here $\widetilde{\mathbb{P}}$ is the synthetic distribution generated by directly applying Laplace mechanism to the entire domain.*

*Proof.* By the definition of Laplace mechanism, we add i.i.d. $\mathrm{Lap}(1/n\epsilon)$ noise to all the $2^d$ choices of $\mathbb{P}(x_1, \cdots, x_d)$ and then normalize them. Then, Theorem 4.3 can be proved similarly as Lemma 6.1 with $m = 2^d$. □

## 4.2 UTILITY ERRORS

In real world practice, the raw training data in supervised learning may contain sensitive information, like personal preference of users. Therefore, it is not allowed to be released to the public. An alternative is to release differentially private synthetic data instead of the raw data. A central problem is the "utility" of synthetic training data, which means the evaluation of synthetic data in downstream tasks. We explain the term utility in detail below.

Consider a dataset $\mathbb{D} = \{x^{(i)}\}_{i=1}^n$ drawn from the domain $\Omega$ with sample size $n$. We denote its corresponding synthetic dataset as $\widehat{\mathbb{D}} = \{\widehat{x}^{(i)}\}_{i=1}^{\widehat{n}}$ of size $\widehat{n}$. We use the empirical risk minimization (ERM) model to capture the supervised learning. Precisely, the ERM estimators for the raw dataset $\mathbb{D}$ and the synthetic data $\widehat{\mathbb{D}}$ are defined as

$$\widehat{\theta} = \arg\min_{\theta \in \mathcal{C}} \frac{1}{n} \sum_{i=1}^n \ell(\theta, x^{(i)}) + \lambda J(\theta),$$

$$\widehat{\theta}_{\text{syn}} = \arg\min_{\theta \in \mathcal{C}} \frac{1}{\widehat{n}} \sum_{i=1}^{\widehat{n}} \ell(\theta, \widehat{x}^{(i)}) + \lambda J(\theta), \tag{2}$$

respectively. Here $\mathcal{C}$ is a convex closed set. We assume the loss function $\ell(\cdot, x)$ is convex on $\mathcal{C}$ and is $L$-Lipschitz in $x$ for some $L \geq 0$. The regularization term $J(\cdot)$ is adopted to prevent over-fitting. The model captures a wide range of applications. For example, given a data point $x^{(i)} = (u_i, v_i) \in \{0,1\}^{d+1}$, by defining the hinge loss $\ell(\theta, x^{(i)}) = (1 - \langle \theta, u_i \rangle \cdot v_i)_+$, we recover the popular support vector machine (SVM) classifier. The loss is $\sqrt{d+1}$-Lipschitz in $\theta$ since $\|x^{(i)}\|_2 \leq \sqrt{d+1}$.

The utility of the synthetic dataset $\widehat{\mathbb{D}}$ measures whether $\widehat{\theta}_{\text{syn}}$ and $\widehat{\theta}$ perform similarly on the prediction task (Esteban et al., 2017). To be specific, the following metric is used to evaluate the utility,

$$U(\widehat{\mathbb{D}}, \mathbb{D}) := \frac{1}{n} \left| R(\widehat{\theta}) - R(\widehat{\theta}_{\text{syn}}) \right|, \tag{3}$$

where $R(\theta) = \sum_{i=1}^n \ell(\theta, x^{(i)})$ is the empirical risk on $\mathbb{D}$ ((Rankin et al., 2020; Hittmeir et al., 2019)). Intuitively, the asymptotic behavior of $U(\widehat{\mathbb{D}}, \mathbb{D})$ is affected by the difference between distributions of synthetic data and true data. This fact is characterized in Theorem 4.4.

We first make the following assumption on the bound of the loss function $\ell(\cdot, \cdot)$, which is quite natural due to its continuity (Bassily et al., 2014).

**Assumption 4.2.** *For any $\theta \in \mathcal{C}$ and any data point $x$ in $\Omega$, we have $|\ell(\theta, x)| \leq 1$.*

**Generating synthetic dataset from PrivBayes.** We still denote the raw dataset as $\mathbb{D}$ and denote $\mathbb{P}$ its empirical distribution. Its corresponding output of PrivBayes is a distribution denoted as $\mathbb{Q}$. To generate the synthetic training data $\widehat{\mathbb{D}}$, we draw $\widehat{n}$ i.i.d. samples from $\mathbb{Q}$. The corresponding empirical distribution is denoted as $\widehat{\mathbb{Q}}$.

With the above preparation, we are now ready to state our result that characterizes the utility of PrivBayes.

**Theorem 4.4.** *If Assumption 4.1 and Assumption 4.2 hold and the raw dataset $\mathbb{D}$ is Boolean, then we have*

$$U(\widehat{\mathbb{D}}, \mathbb{D}) \leq C(\lambda) + C_1 \|\mathbb{Q} - \mathbb{P}\|_{\text{TV}} + 2\mathcal{R}_\mathcal{C} + \sqrt{\frac{\log \frac{1}{\delta}}{2\widehat{n}}}$$

$$\leq C(\lambda) + C_1 \frac{2^{2k} d^2 (k+1)}{n\epsilon} \ln \frac{2d}{\delta} + 2\mathcal{R}_\mathcal{C} + \sqrt{\frac{\log \frac{1}{\delta}}{2\widehat{n}}}, \tag{4}$$

*with probability at least $1 - \delta$. Here $\mathcal{R}_\mathcal{C}$ is the Rademacher complexity of the function class $\{x \mapsto \ell(\theta, x) \mid \theta \in \mathcal{C}\}$ and $C_1$ is a positive universal constant. The term $C(\lambda)$ is non-negative and vanishes when $\lambda = 0$, namely $C(0) = 0$.*

*Proof.* See Section 6 for a proof sketch. □

**Discussion.** The term $C(\lambda)$ comes from the regularization process. $C(\lambda) = 0$ if no regularization is applied ($\lambda = 0$). In real practice, $\lambda$ is often much smaller than $d^2/n\epsilon$. Therefore $C(\lambda)$ is also relatively small. The Rademacher complexity in equation (4) comes from the sampling process.

Our result then implies that, when the sample size $n$ and $\widehat{n}$ are sufficiently large and the regularization parameter $\lambda$ is sufficiently small, the quality loss caused by the private mechanism is rather small. In other words, private synthetic data generated by PrivBayes performs similarly to raw data in downstream learning tasks.

## 5 LOWER BOUND

In this section, we complete the picture by deriving a lower bound for the TV-distance between synthetic private distribution and the raw data distribution.

**Notations and conventions.** As before, the raw dataset $\mathbb{D}$ is of size $n$ with emprical distribution $\mathbb{P}$. The data domain is denoted as $\Omega$. A synthetic data generator is a randomized algorithm that sends a dataset of size $n$ to a distribution over $\Omega$. We also need the following assumption on the range of the parameters.

**Assumption 5.1.** *We assume that $d/\epsilon \ll n \ll |\Omega|$.*

The first part of this assumption allows a rather wide choice of $\epsilon$ in practice. For instance, in two real datasets ACS (Ruggles et al., 2015) and Adult (Bache & Lichman, 2013), the size $n \approx 40,000$, the dimension $d \approx 40$. Then Assumption 5.1 only requires $\epsilon \geq 1/1,000$. Moreover, the size of $\Omega$ is at least $2^d$, which is clearly much larger than $n$. Therefore, the second part of Assumption 5.1 holds for real world datasets.

We now state Theorem 5.1 that establishes the lower bound for TV-distance.

**Theorem 5.1.** *If Assumption 5.1 holds, then for any synthetic data generator $A(\cdot)$ with privacy budget $\epsilon$, and for any $0 \leq \delta \leq 1/2$, there exists a dataset $\mathbb{D}$ of size $n$, such that*

$$\|A(\mathbb{D}) - \mathbb{P}\|_{\mathrm{TV}} \geq \frac{1}{n\epsilon} \log(\delta|\Omega|)$$

*with probability at least $1 - 2\delta$. Here $\mathbb{P}$ is the empirical distribution of $\mathbb{D}$.*

*Proof.* See Appendix D for a detialed proof. $\qquad\square$

Choosing $\delta = 1/4$ in Theorem 5.1 yields the following corollary.

**Corollary 5.2.** *If $|\Omega| \geq 4\exp(d)$, then for any synthetic data generator $A(\cdot)$ with privacy budget $\epsilon$, there exists a dataset $\mathbb{D}$ of size $n$ such that*

$$\|A(\mathbb{D}) - \mathbb{P}\|_{\mathrm{TV}} \geq \frac{d}{n\epsilon}$$

*with probability at least $1/2$. Here $\mathbb{P}$ is the empirical distribution of $\mathbb{D}$.*

**Discussion and comparison.** Comparison with the upper bound in Theorem 4.1, PrivBayes is sub-optimal up to a $d$ factor. The sub-optimality is caused by the composition property of DP (the dataset is processed $d$ times in a Bayesian network) and the structure of a Bayesian network.

## 6 PROOF SKETCH FOR THE TECHNICAL RESULTS

### 6.1 PROOF SKETCH FOR THEOREM 4.1

We begin with a technical lemma that characterizes the normalization process. See Appendix A for its detailed proof.

**Lemma 6.1.** *For a distribution $(a_1, \cdots, a_m)$, we denote its outcome after adding i.i.d. $\mathrm{Lap}(d/n\epsilon)$ noise and normalizing it as $(b_1, \cdots, b_m)$. Then, for all large $n$ and all $\delta > 0$, it holds that*

$$\max_i |a_i - b_i| \leq \frac{3md}{n\epsilon} \log \frac{m}{\delta},$$

*with probability at least $1 - \delta$.*

Lemma 6.1 characterizes the difference between $\widehat{\mathbb{P}}(x_i, \Pi_i)$ and $\mathbb{P}(x_i, \Pi_i)$. However, we need further analysis to establish the conditional version of Lemma 6.1. To be specific, we need to bound $\left|\widehat{\mathbb{P}}(x_i \mid \Pi_i) - \mathbb{P}(x_i \mid \Pi_i)\right|$. The following result serves for this goal.

**Lemma 6.2.** *Consider two non-negative real vectors* $(a_1, \cdots, a_s)$ *and* $(b_1, \cdots, b_s)$ *(not necessary to be distributions). If, for some* $\beta \geq 0$*, we have*

$$\max_j |a_j - b_j| \leq \beta, \tag{5}$$

*then, for any* $l \in \{1, \cdots, s\}$*, the following result holds.*

$$\left| \frac{a_l}{\sum_{j=1}^s a_j} - \frac{b_l}{\sum_{j=1}^s b_j} \right| \leq \frac{s\beta}{\sum_{j=1}^s b_j}. \tag{6}$$

*Proof.* See Appendix A for a detailed proof. $\square$

Combining Lemma 6.1 and Lemma 6.2, the distance between the conditional distributions is bounded in the following result.

**Lemma 6.3.** *If* $\mathbb{D}$ *is boolean and satisfies Assumption 4.1, then we have*

$$\left|\widehat{\mathbb{P}}(x_i \mid \Pi_i) - \mathbb{P}(x_i \mid \Pi_i)\right| \leq \frac{6d2^k(k+1)}{n\epsilon} \log \frac{2}{\delta} \frac{1}{\mathbb{P}(\Pi_i)}, \tag{7}$$

*with probability at least* $1 - \delta$*, simultaneously for all* $i$ *and all choices of* $(x_i, \Pi_i)$*.*

*Proof.* Setting

$$\begin{cases} m = 2^{k+1}, \\ s = 2, \\ a_1 = \widehat{\mathbb{P}}(1, \Pi_i), \ a_2 = \widehat{\mathbb{P}}(0, \Pi_i), \\ b_1 = \mathbb{P}(1, \Pi_i), \ b_2 = \mathbb{P}(0, \Pi_i), \\ \beta = \frac{3md}{n\epsilon} \log \frac{m}{\delta}, \end{cases}$$

in Lemma 6.1 and Lemma 6.2 concludes the proof. $\square$

To bound the TV-distance, we begin with rewriting it in telescoping series and applying Lemma 6.3. One technical impediment for estimation is the fraction term $1/\mathbb{P}(\Pi_i)$ in (7). To address this challenge, we need to deploy the Bayesian network structure (Assumption 3.1 and Assumption 3.2). Deploying the network structure is quite technical and lengthy, we defer the detail to Appendix A.

## 6.2 PROOF SKETCH FOR THEOREM 4.4

We begin with some notations. The non-regularized estimators trained on $\mathbb{D}$ and $\widehat{\mathbb{D}}$ by an ERM model are denoted as $\theta^*$ and $\theta^*_{\text{syn}}$. Formally, we define

$$\theta^* := \arg\min_{\theta \in \mathcal{C}} \frac{1}{n} \sum_{i=1}^n \ell(\theta, x^{(i)}),$$

$$\theta^*_{\text{syn}} := \arg\min_{\theta \in \mathcal{C}} \frac{1}{\widehat{n}} \sum_{i=1}^{\widehat{n}} \ell(\theta, \widehat{x}^{(i)}).$$

We further define the prediction risk with respect to a certain distribution. For any distribution on $\Omega$, denoted as $P$, and any $\theta \in \mathcal{C}$ we define

$$R(\theta, P) := \sum_{x \in \Omega} \ell(\theta, x) P(x) \tag{8}$$

as the prediction risk with respect to $P$. Then $R(\cdot)$ in (3) is equal to $R(\cdot, \mathbb{P})$.

We are now ready to sketch the proof. The most important step of the proof is to decompose the utility in (3 into the following seven terms

$$
\begin{aligned}
U(\widehat{\mathbb{Q}}, \mathbb{P}) \leq \\
\underbrace{\left| R(\widehat{\theta}_{\mathrm{syn}}, \mathbb{P}) - R(\widehat{\theta}_{\mathrm{syn}}, \mathbb{Q}) \right|}_{\text{term (i)}} + \underbrace{\left| R(\widehat{\theta}_{\mathrm{syn}}, \mathbb{Q}) - R(\widehat{\theta}_{\mathrm{syn}}, \widehat{\mathbb{Q}}) \right|}_{\text{term (ii)}} + \underbrace{\left| R(\widehat{\theta}_{\mathrm{syn}}, \widehat{\mathbb{Q}}) - R(\theta_{\mathrm{syn}}^*, \widehat{\mathbb{Q}}) \right|}_{\text{term (iii)}} \\
+ \underbrace{\left| R(\theta_{\mathrm{syn}}^*, \widehat{\mathbb{Q}}) - R(\theta^*, \widehat{\mathbb{Q}}) \right|}_{\text{term (iv)}} + \underbrace{\left| R(\theta^*, \widehat{\mathbb{Q}}) - R(\theta^*, \mathbb{Q}) \right|}_{\text{term (v)}} + \underbrace{\left| R(\theta^*, \mathbb{Q}) - R(\theta^*, \mathbb{P}) \right|}_{\text{term (vi)}} \\
+ \underbrace{\left| R(\theta^*, \mathbb{P}) - R(\widehat{\theta}, \mathbb{P}) \right|}_{\text{term (vii)}} .
\end{aligned} \tag{9}
$$

Recall that $\mathbb{Q}$ is the output of PrivBayes and $\widehat{\mathbb{Q}}$ is the empirical distribution of the $\widehat{n}$ samples drawn independently from $\mathbb{Q}$. Here term (i) and term (vi) come from the difference between the synthetic distribution $\mathbb{Q}$ and the raw distribution $\mathbb{P}$. They can be bounded above by the distance between the synthetic distribution and the raw one. Term (ii) and term (v) come from sampling and are bounded by classical Rademacher method. Term (iii) and term (vii) are derived from the regularization process. They combine to be the $C(\lambda)$ term. Bounding term (iv), however, is more tricky and requires more detailed analysis. We discuss each group in detail in Appendix C.

## 7 DISCUSSIONS AND FUTURE TOPICS

We establish perhaps the first statistical analysis for the accuracy and utility of Bayesian network-based data synthesis algorithms. We also derive a lower bound for the accuracy to complete the picture. Compared with the lower bound, the accuracy bound we achieve is sub-optimal up to a $d$ factor. One way to improve the accuracy is to reduce the effects of random noise in releasing the synthetic data through some post-processing procedures. However, it is still quite challenging to develop a practical algorithm based on this idea, and we leave it for future work.

## ACKNOWLEDGMENTS

We appreciate Prof. Ninghui Li and Dr. Zitao Li for their discussions about the background and applications of marignal-based data synthesis methods, which motivates us to study the corresponding theory. This research is supported by the Office of Naval Research [ONR N00014-22-1-2680] and the National Science Foundation [NSF – SCALE MoDL (2134209)].

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
