# OpenReview forum: "Statistical Theory of Differentially Private Marginal-based Data Synthesis Algorithms"
_ICLR.cc/2023/Conference — ICLR 2023 poster_

### Official Review · Reviewer_BqvB · 2022-10-24

**Confidence:** 3
**Correctness:** 4
**Technical Novelty And Significance:** 4
**Empirical Novelty And Significance:** Not applicable
**Recommendation:** 6

**Clarity, Quality, Novelty And Reproducibility:**

The paper is clear and well-written. To the best of my knowledge, this statistical analysis of PrivBayes is new.

**Strength And Weaknesses:**

Strengths
- The paper tackles an interesting problem, synthetic data generation under differential privacy
- The paper is clearly written and relatively straightforward to follow
- The bounds provide an interesting picture on the dependency to various factors (size of training data n, dimension d, 'breadth' of the model k, budget epsilon, etc.)

Weaknesses
- the upper bound seems rather loose. For example, theorem 4.1 depends exponentially on k. If we plug n=40k, d=40 and epsilon=10 which are generarous estimates, we get 12 d^2/(n*epsilon) ~ 0.05 so with k=3 the bounds becomes vacuous (and that's without looking at the log factor).

**Summary Of The Paper:**

The paper tackles the general problem of synthetic data estimation under differential privacy (DP). Specifically, the authors prove upper and lower bounds on DP estimation of PrivBayes, a marginal-based Bayesian network synthetic data generation method. More specifically, the authors prove upper-bounds on the total variation and L2 distance between the estimated distribution and the true distribution under the assumption that the raw data is generated from a graph with causal edges and bounded number of parent nodes for each node. The authors also show utility upper bounds, as well as total variation lower bounds.

**Summary Of The Review:**

Overall I think the paper provides interesting bounds on a classical mechanism for synthetic data generation, PrivBayes. I am a bit on the fence for a general recommendation as I am not 100% familiar with the synthetic data literature but I lean towards borderline+.

---

> ### Author Response · Authors · 2022-11-11
> **Response to the Reviewer**
>
> The authors thank for the reviewer's comments. These comments are all valuable and very helpful for revising and improving the paper, as well as the important guiding significance to our research. We have studied the comments carefully. The responses to the reviewer's comments are as follows:
>
> Q: The bound is loose and exponential in $k$?
>
> A: The $\exp(k)$ factor is reasonable in the following sense. The $\mathrm{TV}$ distance is integrating difference of densities on the total domain. Therefore, the $\mathrm{TV}$ distance between $\widehat{{P}}(x_i,\Pi_i)$ and ${P}(x_i,\Pi_i)$ is the summation of $O(e^k)$ positive terms, where $O(e^k)$ is the domain size. In this sense, it is reasonable that the bound is exponential in $k$. This is also the advantage of using Bayesian networks since the bound can be exponential in $d$ without using BN (Theorem 4.3).
>
> Moreover, PrivBayes has a good performance on sparse data with small $k$. However, when $k$ is large, there might be other marginal-based methods that outperform the PrivBayes, which is in line with our theory.
>
> The bound is rather loose and we will tighten it in our future study. As we mentioned in the paper, one $d$ in the upper bound is from the composition properties of DP.  In our future work, we will consider the Gaussian mechanism using R'enyi DP or Gaussian DP, which will lead to tighter composition bounds (cf., Corollary 1 in [Mironov, 2017, Renyi Differential Privacy]). Technically, the proof of the tighter bound adopts R'enyi DP or Gaussian DP and is much more involved than the proof of the $\epsilon$-DP case in this paper. Since both the DP tools and the technical details are different from this paper, we postpone the detailed discussion in our follow-up papers.

---

### Official Review · Reviewer_mQvM · 2022-10-24

**Confidence:** 3
**Correctness:** 3
**Technical Novelty And Significance:** 2
**Empirical Novelty And Significance:** 2
**Recommendation:** 6

**Clarity, Quality, Novelty And Reproducibility:**

The paper is clearly written, the quality of the empirical work seems fine but there is a lack of adequate theoretical developments. The algorithm may have some elements of novelty, and the results look reproducible.


**Strength And Weaknesses:**

Strengths:


Authors have a good grasp of some of the challenges of differentially private computations in the context of the paper. The paper is generally well-written. The framework of using a BN for distribution representation is sensible n many applications, hence there is merit to the problem under study.


Weakness:

The paper only focuses on Boolean data. There is an aside claim that it can be generalized to any categorical data with finite support, but this extension is not clear from the paper. Also, it does not seem that the main steps would extend easily without considerable problems related to scaling of the computations. It is not clear to me whether the BN is estimated from the data, or is considered known. If it is estimated from the data, is the estimation algorithm differentially private?


**Summary Of The Paper:**


Authors consider the case where the synthetic data distribution can be represented by a Bayesian network, while the raw data distribution is approximated by a collection of low-dimensional marginals. Differential privacy (DP) is guaranteed by introducing random noise to each low-dimensional marginal distribution. Authors study DP data synthesis algorithms based on Bayesian networks (BN) from a statistical perspective.


**Summary Of The Review:**

Restriction to just Boolean data is a major limitation. Other than that, there are positive points about this paper.


Revised report: I had given a score of 5 (marginally below acceptance) initially.  Given the clear answers provided by the authors, especially to the main question I had, i am happy to change my recommendation and now recommend acceptance.

---

> ### Author Response · Authors · 2022-11-11
> **Response to the Reviewer**
>
> The authors thank for the reviewer's comments. These comments are all valuable and very helpful for revising and improving the paper, as well as the important guiding significance to our research. We have studied the comments carefully. The responses to the reviewer's comments are as follows:
>
> Q1: The paper only focuses on Boolean data.
>
> A1: Our work can be generalized as follows.
>
> For general tabular data, we assume the data domain $\Omega$ is a subset of $\\{1,2,\ldots,p\\}^d$. Note that Lemma 6.1 and 6.2 holds for general tabular data. Therefore, by substituting $m=p^{k+1}$ in Lemma 6.1 and $s=p$ in Lemma 6.2, we generalize Lemma 6.3 to general tabular case as follows.
>
>
> **Lemma 6.3 (Generalized)**:
>
> Denote the raw dataset as $D$. If the corresponding data domain, denoted as $\Omega$, is a subset of $\\{1,2,\ldots,p\\}^d$. Moreover ${D}$ satisfies Assumption 4.1, then we have
> \begin{equation}
> |{\widehat{{P}}(x_i\mid \Pi_i)-{P}(x_i\mid \Pi_i)}|
> \leq \frac{6 d p^k(k+1)}{n \epsilon}\log \frac{pd}{\delta}\frac{1}{{P}(\Pi_i)},
> \end{equation}
> with probability at least $1-\delta$, simultaneously for all $i$ and all choices of $(x_i,\Pi_i)$.
>
> By adopting Lemma 6.3 (Generalized) and repeating the argument of Theorem 4.1, we can show that the TV distance is bounded as
> \begin{equation}
> \left\lVert{\widehat{{P}}-{P}}\right\rVert_{\mathrm{TV}}
> \leq C\frac{d^2 p^{2k} (k+1)}{n \epsilon}\log \frac{pd}{\delta},
> \end{equation}
> with probability at least $1-\delta$, where $C$ is a universal constant. Therefore, we establish our result for general tabular case.
>
> For more general cases like continuous data with domain $[M_1,M_2]^d$ which is a continuous interval, the density estimation methods are different from the discrete case.
> We believe that our analysis in the paper can be used for continuous data using the idea from [Wasserman & Zhou, 2010, A statistical framework for differential privacy] where continuous distributions are considered.
> Since the technical proof is nontrivial, we will study the continuous case in our future study.
>
>
> Q2: Is the BN estimated from the real data? If so, is the estimation algorithm DP?
>
> A2: Yes, it is estimated by the algorithm PrivBayes. [Zhang et.al., 2017, PrivBayes] adopts a greedy algorithm to estimate the structure of the BN and shows that it enjoys DP.

---

### Official Review · Reviewer_sYaP · 2022-10-25

**Confidence:** 3
**Clarity, Quality, Novelty And Reproducibility:** Yes
**Correctness:** 4
**Technical Novelty And Significance:** 3
**Empirical Novelty And Significance:** Not applicable
**Recommendation:** 6

**Strength And Weaknesses:**

The authors derived three novel bounds which has not been carefully studied before. I found the proofs are compelling and the well-conveyed. The authors deliver the proofs clearly and they bound the utility errors by neatly splitting the errors into several parts.

However, the authors only derived new bounds under total variation distance and L^2 distance. Perhaps more statistical distances should be employed to evaluate the synthetic data for higher completeness of the paper.



**Summary Of The Paper:**

This paper analyzes the accuracy of PrivBayes by giving upper bounds on the distances between real and synthetic data and on utility errors of synthetic data from downstream supervised learning tasks.  The authors also proved the lower bound for total variation distance for \epsilon-DP synthetic data generator.

**Summary Of The Review:**

In general, the findings are interesting, and the work is well-supported and well-organized. Although the results leave rooms for further improvement, I would recommend this paper to be accepted.

---

> ### Author Response · Authors · 2022-11-11
> **Response to the Reviewer**
>
> The authors thank for the reviewer's comments. These comments are all valuable and very helpful for revising and improving the paper, as well as the important guiding significance to our research. We have studied the comments carefully. The responses to the reviewer's comments are as follows:
>
> Q: Perhaps more statistical distances should be employed to evaluate the synthetic data for higher completeness of the paper.
>
> The reason for focusing on $\mathrm{TV}$ distance is that it is closely related to the utility of downstream learning task (see Section 4.2). Meanwhile, discussing $L^2$ error is a natural choice for the post-processing $L^2$-projection.
>
> There are some other statistical distances that we are interested in. One is the $\mathrm{KS}$  distance (Kolmogorov–Smirnov). It is smaller than the $\mathrm{TV}$ distance, so Theorem 4.1 yields a bound for $\mathrm{KS}$ distance. The second choice is the $\mathrm{KL}$ divergence. The definition of $\mathrm{KL}({{P}}\mid \widehat{{P}})$ requires that ${{{P}}}$ is absolutely continuous w.r.t $\widehat{{P}}$. However, this condition may fail for Marginal-Based methods. Precisely, for some $a\in\Omega$ such that ${P}(a)>0$, denote the Laplace noise added to ${P}(a)$ as $v$. Then the normalization process implies that $\widehat{{P}}(a)=0$ whenever ${P}(a)+v<0$. Therefore, ${{{P}}}$ is $\textbf{not}$ absolutely continuous w.r.t $\widehat{{P}}$ with a positive probability.

---

### Official Review · Reviewer_PzT1 · 2022-11-04

**Confidence:** 3
**Correctness:** 4
**Technical Novelty And Significance:** 3
**Empirical Novelty And Significance:** Not applicable
**Recommendation:** 6

**Clarity, Quality, Novelty And Reproducibility:**

The paper is pretty clear on its contributions with proof sketches for the main results sprinkled throughout the main body. The problem setting is well motivated and the novelty is in deriving rigorous guarantees.


**Strength And Weaknesses:**

Pros:
     (A) Establishes the quality of the synthetic datasets by providing concrete distance upper-bounds on the generated datasets which are O(d^2 2^k/n) ignoring log factors and setting epsilon to 1. Similarly, utility of the datasets is provided for the downstream tasks.
     (B) Lower bounds are also provided for the hypercube setting and is O(d/n). The suboptimality arises due to the composition component of DP.
Cons:

   (i) While the theoretical guarantees that are shown in the paper are nice, it is not clear how tight they are? For typical values of k, d and n the values seem
to be way larger than 1 and not entirely useful?

   (ii) Experiments showing the tightness of the lower and upper bounds on real-world datasets would have been helpful in understanding the strengths and limitations of the theory.


**Summary Of The Paper:**

The paper studies the problem of Bayesian structure based synthetic data generation and its theoretical guarantees with respect to (probability) distance from the real data. Also, lower bounds for the TV distances between the two distributions is provided and differ by a factor of dimension d.

**Summary Of The Review:**

Overall, the paper studies an important problem setting by providing rigorous theoretical guarantees for synthetic data generation. I have some questions which I hope to discuss with the authors.

---

> ### Author Response · Authors · 2022-11-11
> **Response to the Reviewer**
>
> The authors thank for the reviewer's comments. These comments are all valuable and very helpful for revising and improving the paper, as well as the important guiding significance to our research. We have studied the comments carefully. The responses to the reviewer's comments are as follows:
>
> Q1: How tight the bound is? For typical values of $k$, $d$ and $n$ the bound may be vacuous?
>
> A1: For example for the data sets [Integrated
> public use microdata series: Version 6.0] , we have $n\approx 40,000$ and $d\approx 20$. For $k\leq2$, the error bounds for typical values of $\epsilon$ are listed in the table below. The bounds are not vacuous. For larger $k$, the bound may be vacuous. This is due to the fact that PrivBayes has good performance on sparse data with small $k$. However, when $k$ is large, there might be other marginal-based methods that outperform the PrivBayes, which is in line with our theory.
>
>
>
>  Value of $\epsilon$:   1   |      2   |    4   |    8
>  $\mathrm{TV}$ error:  0.16  |  0.08 | 0.04 | 0.02
>  $L^2$ error:  0.04 |  0.02  |   0.01 |   0.005
>
>
> Considering that the lower bound is approximately $1/2000$, the upper bound is rather loose.
> As we mentioned in the paper, one $d$ in the upper bound is from the composition properties of DP. In our further work, we will consider the Gaussian mechanism using R\'enyi DP or Gaussian DP, which will lead to tighter composition bounds (cf., Corollary 1 in [Mironov, 2017, Renyi Differential Privacy]). Technically, the proof of the tighter bound adopts R\'enyi DP or Gaussian DP and is much more involved than the proof of the $\epsilon$-DP case in this paper.
> Since both the DP tools and the technical details are different from this paper, we postpone the detailed discussion in our follow-up papers.
>
> Q2: Related experiments will be helpful.
>
> A2: Due to the time limit, we will evaluate the upper bound empirically in our future study.
> The PrivBayes algorithm is widely used in many existing libraries such as the DataSynthesizer [Ping et.al., 2017, DataSynthesizer] and The Synthetic Data Vault [Patki et.al., 2016, The Synthetic Data Vault].
> We plan to use the two libraries above to generate synthetic data from both real datasets and toy Boolean datasets.
>
>  Thank you very much for your interest. We are open to discussion in further comments.

---

### Decision · Program_Chairs · 2023-01-20

**Decision:**

Accept: poster

**Justification For Why Not Higher Score:**

The contributions are above the bar of acceptance but they are not of a profound impact and/or wide interest to merit an oral or spotlight slot.

**Justification For Why Not Lower Score:**

The paper makes clear contributions and is above the bar of acceptance.

**Metareview: Summary, Strengths And Weaknesses:**

The paper studies Bayesian differentially private (DP) synthetic data generation. In this work, synthetic data is generated via Bayesian Network (BN) while raw data is approximated by a set of low-dimensional marginals.

Generating synthetic data in a privacy preserving manner is an important and timely problem with far-reaching practical impacts. The authors provide a set of interesting results. They give formal accuracy guarantees for DP BN-based algorithms in terms of upper bounds on the Total Variation (TV) and L_2 distance between the distributions of the raw and synthetic data. The authors also establish a lower bound on TV-based accuracy of any pure DP algorithm for synthetic data generation. The paper is also well written. There is a general consensus among the reviewers that the formal results of this work are interesting and useful.

**Note From Pc:**

if the above contains the word "oral" or "spotlight" please see: "oral" presentation means -> notable-top-5% and "spotlight" means -> notable-top-25%. As stated in our emails, we are disassociating presentation type from AC recommendations